# Informality, Social Citizenship, and Wellbeing among Migrant Workers in Costa Rica in the Context of COVID-19

**DOI:** 10.3390/ijerph19106224

**Published:** 2022-05-20

**Authors:** Mathieu J. P. Poirier, Douglas Barraza, C. Susana Caxaj, Ana María Martínez, Julie Hard, Felipe Montoya

**Affiliations:** 1School of Global Health, Faculty of Health, York University, Toronto, ON M3J 1P3, Canada; 2Global Strategy Lab, Dahdaleh Institute for Global Health Research, York University, Toronto, ON M3J 1P3, Canada; 3Health Section, Central American Institute for Studies on Toxic Substances, Universidad Nacional, Heredia 41101, Costa Rica; dbarraza@utn.ac.cr; 4Coordinación de Docencia, Universidad Técnica Nacional, San Carlos 21001, Costa Rica; 5School of Nursing, University of Western Ontario, London, ON N6A 5B9, Canada; scaxaj@uwo.ca; 6York International & Faculty of Environmental and Urban Change, York University, Toronto, ON M3J 1P3, Canada; ana_martinez@edu.yorku.ca; 7Faculty of Health, York University, Toronto, ON M3J 1P3, Canada; jlhard@yorku.ca; 8Faculty of Environmental and Urban Change, York University, Toronto, ON M3J 1P3, Canada; fmontoya@yorku.ca

**Keywords:** migrant health, social determinants of health, Costa Rica, Nicaragua, COVID-19, grounded theory, informal work, welfare states, social citizenship, documentation

## Abstract

Costa Rica is home to 557,000 migrants, whose disproportionate exposure to precarious, dangerous, and informal work has resulted in persistent inequities in health and wellbeing in the midst of the COVID-19 pandemic. We used a novel multimodal grounded approach synthesizing documentary film, experiential education, and academic research to explore socioecological wellbeing among Nicaraguan migrant workers in Costa Rica. Participants pointed to the COVID-19 pandemic as exacerbating the underlying conditions of vulnerability, such as precarity and informality, dangerous working conditions, social and systemic discrimination, and additional burdens faced by women. However, the narrative that emerged most consistently in shaping migrants’ experience of marginalization were challenges in obtaining documentation—both in the form of legal residency and health insurance coverage. Our results demonstrate that, in spite of Costa Rica’s acclaimed social welfare policies, migrant workers continue to face exclusion due to administrative, social, and financial barriers. These findings paint a rich picture of how multiple intersections of precarious, informal, and dangerous working conditions; social and systemic discrimination; gendered occupational challenges; and access to legal residency and health insurance coverage combine to prevent the full achievement of a shared minimum standard of social and economic security for migrant workers in Costa Rica.

## 1. Introduction

Since the turn of the 20th century, immigration to Costa Rica has been spurred by its relative political and social stability, as well as its level of economic development, particularly when compared to other countries in the region that have faced various social, economic, and political crises [1]. At the onset of the 1980s, immigration accelerated with the arrival of tens of thousands of refugees fleeing armed conflict and economic turmoil in Nicaragua [2]. As of 2020, there were approximately 557,000 documented migrants living in Costa Rica, representing 10% of the total population, although the true figure is likely higher than 15% when accounting for undocumented migrants [3].

Nicaragua is, by far, the most common country of origin with 367,984 migrants (66.1% of all migrants), followed by Colombia with 28,887 (5.2%) and the United States with 28,731 (5.1%) [1]. A more recent survey of six Central American countries found that Nicaraguans accounted for 86.7% of self-identified migrants within the region, with Costa Rica identified as the destination country for 81.5% of respondents [3]. Although the COVID-19 pandemic slowed the rate of migration to the country, the Costa Rican economy continues to rely on primarily Nicaraguan migrant labour, highlighting the importance of understanding the dynamics of south–south migration [4].

In spite of these economic contributions, many migrant workers are informally employed, working without a contract and without equal access to the vaunted Costa Rican welfare state [5]. This situation produced a dynamic of vulnerability in which workers not only become dependent on the good will of their employers but also may never qualify to receive a retirement pension at the end of their working lives. 

According to the Costa Rican Ministry of Labor and Social Security (MTSS), migrants are allowed to work in Costa Rica only if they are in possession of a resident work permit, temporary work permit, or are a recognized refugee or asylum seeker [6]. If a migrant worker fulfills one of these requirements, they are provided with access to healthcare through the Costa Rican Social Security Fund (CCSS) and a pension upon retirement. Although these legal pathways to work, healthcare, and pensions are available, access to these fundamental government services has been worsening in recent years.

In 2011 and 2018, the Central American Program on Work and Health (SALTRA) in conjunction with the School of Public Health of the University of Texas and Pompeu Fabra University conducted the first and the second rounds of the Central American Surveys on Working Conditions in Guatemala, El Salvador, Honduras, Nicaragua, Costa Rica and Panama [7]. In this seven-year span, the proportion of respondents who reported not having access to social security rose from 68% to 72%. The figures for Costa Rica were even more concerning than the Central American average, with the proportion not having access to social security rising from 34% in 2011 to 59% in 2018, and pandemic-related impacts likely further reducing access since that time.

This increase in informal work over the last decade has not only prevented access to government services but may also be affecting exposure to labour risks and hazards. A recent report on the prevalence of labor risk agents among the Central American labour force (informal and formal) presented in Table 1 demonstrates the pervasive exposure to noise, toxic chemical substances, repetitive movements, and rapid pace, all while lacking access to labor risk-prevention services and union health and safety services.

Echoing experiences in the global north [8], migrant workers often bore the brunt of the economic, social, and health impacts of the pandemic in Costa Rica. Nicaraguan migrants were disproportionately affected by workplace COVID-19 outbreaks in the early months of the pandemic, resulting in some prominent voices laying blame on migrants themselves [9,10,11]. These tensions resulted in xenophobic attacks and intimidation of migrants, sometimes motivated by the Nicaraguan government’s anemic public health response [12,13]. This underscores the importance of understanding the challenges faced by migrant workers in the country as well as the importance of identifying opportunities to improve the social determinants of health in the midst of the COVID-19 pandemic with a focus on the understudied dynamics of south–south migration and social integration.

This study brings light to several challenges facing Nicaraguan migrants in Costa Rica, which emerged from our grounded theory approach on topics including precarity and informality, dangerous working conditions, social and systemic discrimination, and the additional burdens faced by women. However, the narrative that emerged most consistently in shaping migrants’ experience of marginalization were challenges in obtaining documentation—both in the form of legal residency and health insurance coverage. 

Finally, rather than being identified as a direct cause of ill health, the COVID-19 pandemic underscored and exacerbated the underlying conditions of vulnerability that study participants had been dealing with for years. The notion of COVID-19 as an exacerbating, rather than novel, challenge to the health of migrant agricultural workers has been documented in various countries [14,15]. Through a novel multimodal grounded approach, we give voice to Nicaraguan migrant experiences of marginalization in Costa Rica with the aim of illustrating shared challenges and opportunities for action to address these underlying determinants of health in countries around the world.

## 2. Materials and Methods

The Grounded project, a joint experiential education and research initiative of the Faculty of Health and the Faculty of Environmental and Urban Change of York University, was named for both our novel multimodal grounded approach as well as the grounding of travel in response to the COVID-19 pandemic. The aim of this study was to bring together an interdisciplinary team of Canadian academic researchers with Costa Rican experts, local stakeholders, and community members to collectively explore and further the understanding of the threats and opportunities, difficulties and potentials, fears, and aspirations for socioecological wellbeing in the global south. 

Our grounded approach was guided by the principle of allowing migrant workers in Costa Rica to tell their own stories [16], which we strove to represent meaningfully in the form of a documentary film, experiential education opportunities, and academic research. Through a process of continued engagement and the employment of multiple methods, our research was informed by a generic grounded theory approach, which enabled theory-sensitive exploration alongside the coproduction of knowledge with participants [16]. Participatory principles of relational meaning-making, and storytelling through locally appropriate platforms [17] further guided the mediums and activities chosen to both disseminate and elaborate on the preliminary research findings.

### 2.1. Data Collection

During the first quarter of 2021 we performed 13 in-depth interviews of Nicaraguan migrants in their home settings, ranging from 45 min to two hours each, as well as two in-depth interviews with an expert and an activist, each in their work settings. The interviews of Nicaraguan migrants took place in two regions of the country. In the Northern Region near Costa Rica’s border with Nicaragua (Ciudad Quesada, Muelle, and Boca de Arenal), we interviewed four Nicaraguan migrants, and in the Atlantic Region (Guápiles), we performed nine interviews. The expert and activist were interviewed in Ciudad Quesada and La Carpio of San José, respectively (Table 2). The semi-structured interviews, whose guiding questions focused on perceptions and obstacles to the achievement of wellbeing, were developed in consultation with academic leads and students as part of the international experiential program. 

The interviews were recorded with audiovisual equipment (Sony Alpha III video camera and lapel microphones) and uploaded to a virtual platform for translation and analysis of the Grounded project research and education team. The research team in the field responsible for data collection was composed of an interviewer, a videographer, and a field producer who engaged directly with community stakeholders, while the remaining research team used virtual platforms for data analysis and to access interview materials for creation of documentary films based on emergent themes.

Ethics approval was provided by the Human Participants Review Sub-Committee in the York University’s Ethics Review Board (Certificate #2020-374), which conforms to the standards of the Canadian Tri-Council Research Ethics guidelines. Written and oral informed consent forms were translated into Spanish and given to each participant before the interviews were conducted, and protocols to prevent the transmission of COVID-19 were followed at all stages of research.

### 2.2. Data Analysis

Following an inductive approach, where a series of open-ended questions allowed for conversations to emerge naturally regarding interviewee experiences in pursuing their wellbeing as Nicaraguan migrants in Costa Rica, we analyzed data through iterative stages of recording, transcription, translation, initial coding, focused coding, and theoretical sampling [19]. The qualitative data collected in recorded interviews were first transcribed in Spanish and translated to English through a collaborative process, matching students and professors trained in translation, health, and environmental science at the Universidad Técnica Nacional (UTN) in San Carlos, Costa Rica with students and professors at York University (YU) in Toronto Canada.

Initial and focused coding consisted of detailed note-taking on emergent themes, noteworthy word choices, and contextual visual information using a constant comparative approach. This was followed by thematic coding, in which coders separated quotes into emergent themes, further refining the themes according to the underlying data. Finally, theoretical sampling was conducted collaboratively through weekly meetings from January to May 2021, with all authors participating, in addition to all Grounded project students, staff, and faculty.

Once theoretical saturation was reached, illustrative quotes from eight of the 15 participants were extracted and ordered according to the emergent themes. Information that was not fully captured through the illustrative quotes is summarized according to each emergent theme. Thematic analysis informed the creation of the structure for sections or “chapters” of the documentary film along with topics for independent student inquiry and research in response to the grounded research approach. Unique to our approach was the simultaneous analysis of qualitative data and the construction of an accompanying open-access documentary film titled *More Than Migrants* [20].

Finally, our emergent themes were member-checked and validated in the form of three documentary screenings. A first invitation was extended to all participants and their guests in the Northern Region to attend the film screening, which was held on 4 June 2021 at Universidad Técnica Nacional in Ciudad Quesada. This screening was attended by 20 in-person participants and was also live-streamed on Facebook with the participation of virtual visitors followed by an opportunity for participants to express reactions to the film. 

A second invitation was extended to participants in the Atlantic Region held in a Community Centre in *Los Precarios de Jiménez* (the slums of Jiménez) with the participation of 20 attendees, including the interviewed participants as well as their invited guests. Reactions voiced during both screening events as well as in the comment section of YouTube were overwhelmingly positive. One sentiment that was voiced by several participants, alluded to the importance of having seen other Nicaraguan migrants tell their stories, allowing them to feel part of a common experience shared by others.

A third screening was held virtually through YU on 10 June 2021 and was attended by over 150 participants. This screening was followed by a panel discussion with Grounded project team members (MJP, DB, and FM) and a migrant health scholar (CSC), which allowed for direct questions from viewers in Canada, Costa Rica, and other countries. 

Using a virtual format for delivery and engagement, participants could opt to remain anonymous to deliver critical feedback or offer opposing viewpoints to the themes identified. Nevertheless, participants may have refrained from sharing critical feedback during in-person screenings, and anonymous interview methods may have resulted in a different set of emergent themes. The results that follow are a product of our novel multimodal grounded approach and collaborative data collection, analysis, member checking, and validation, beginning with the COVID-19 pandemic as our port of entry.

## 3. Results

### 3.1. “COVID Has Come to Mark Us All”

The COVID-19 pandemic was overwhelmingly perceived as a source of amplification and deterioration of risks, challenges, and sources of vulnerability that long preceded the pandemic, rather than being an immediate health concern. Without access to health insurance, one participant could not afford to visit family in Nicaragua because of the COVID-19 testing and documentation requirements needed to return to Costa Rica. Several participants reported layoffs at their workplace, a lack of new job opportunities, and disruptions to education:


*María: I was taking a pharmacy assistant course, but after I left my job, I couldn’t [afford] the course anymore. Later I started another job, as a domestic [worker] but with older adults. I wanted to keep taking the course, but the salary wasn’t enough. So, I couldn’t continue with my studies, and the work continued, but then with the pandemic, the work ended as well. So now I’m out of work and out of school.*


With the pandemic resulting in the temporary closure of certain government offices and a slowdown in the pace of processing documents, some participants reported indefinite waits until their health insurance applications would be approved. There was general agreement that the pandemic had disproportionately affected women, young people, families with young children, and Nicaraguan migrants.


*Elizabeth: The pandemic affected Nicaraguans much more, well I speak for us, right. Because we’re the ones on the streets. We’re selling bananas, selling masks, selling it like, pulseándola (i.e., pushing forward), as we Nicaraguans say, because it is quite a difficult job. Even more so for those who have children. And with childcare and everything else, you end up earning nothing.*


Despite the pandemic’s clear impacts on their wellbeing, participants repeatedly directed attention towards the struggles of trying to establish themselves in Costa Rica. Of these, the first obstacle to overcome—and one of the most difficult—was securing a work permit or documentation of permanent residence.

### 3.2. Fighting for a Life of One’s Own

Participants were composed of a mixture of documented and undocumented migrants, and the legal barriers to obtaining paperwork varied between those that migrated decades ago and those that migrated more recently. Documents were consistently described as being expensive, difficult to obtain and maintain, and easily revoked upon failure to apply for renewal. These challenges were compounded by decreased government capacity to provide documentation to applicants in a timely manner due to the pandemic. One individual described the hardships they endured trying to access legal status:


*Miguel: And as soon as there was the first amnesty, I slept on the street to see if I could get it. The first day I didn’t get anything because they were serving about 200 people. When I got there in line there were more than 500 people. So, what did we have to do? Sleep in the street, enduring the sun, as well as rain, cold, hunger… I fought hard to get a document that was legal.*


Participants had to take time away from work to face the uncertain prospect of long lines, lengthy wait times, and complex paperwork, all of which resulted in delays of up to ten years before securing the paperwork. Lack of documentation was reported as causing anxiety and mistrust of government officials. For instance, participants voiced suspicion that workplace safety checks conducted by the Department of Migration were in fact intended to deport undocumented workers. In some cases, migrant workers also had to overcome challenges related to low levels of literacy, disease and disability, and tragic loss of life. One participant discussed the losses she endured in Nicaragua that drove her to find a new life in Costa Rica:


*Silvia: Due to many events, my children were left without a father… From that time on, I no longer had stability in Nicaragua, or in anything… I came from there with three children. So, I was wandering from one place to another, and I was renting [housing], and I went through an experience where they fumigated, the poison was very strong, my child died, and I almost died too… Someone whose child has died is like ripping out their heart. I despaired and went to my former mother-in-law, and she told me that here in Costa Rica one could earn money, that one could quickly get a plot of land and have a life of one’s own.*


Even as migration was commonly identified as a difficult but necessary option for obtaining an acceptable standard of living, it was also clear that one’s ability to migrate was heavily influenced by multiple intersecting identities and vulnerabilities ranging from age, gender, poverty, and Indigenous heritage. One participant took pride in his ability to overcome some of these barriers to migration:


*Miguel: Look, from the moment I came here my aspiration has always been to have a stable place to be. Something that belongs to me. And what I’m going to get, my idea has always been to get it through my own sweat and sacrifice. And set a good example to my family, my sons, my daughters, right, and to the neighbors too. That they see that you can do things, you know, fighting without harming anyone. And not wanting to take what is not yours. But rather, to get ahead through your own work… I’m from the department of León. Sutiaba neighborhood. Pure Indian (i.e., Indigenous). Here you have me. Here I am. Look how far I’ve walked already.*


### 3.3. “They Call You ‘Nica’”

Even with documentation, many participants continued to face many forms of social discrimination in the north of the country. Participants migrated from Nicaragua for many reasons, ranging from pursuing economic opportunity, to fleeing war and interpersonal violence; however, regardless of these differences, the experiences of being denied rental housing, being fired from work, or being the victims of discriminatory verbal harassment were mentioned frequently. One participant identified trade unions as a rare source of social support and employment security:


*Santiago: [They would say], ‘For them to get involved [with trade unions], they have to be Nicaraguan.’ But no! It’s just that I had no other choice. Because I have a family to support. I was forced to do this. I didn’t want to do this, but I was forced to do it. And you know why? Because if, as they say there is no discrimination, then why did I get fired if I’m giving a hundred percent effort?*


Some participants identified the root of their experiences of discrimination with competition for employment in the agricultural sector with Costa Rican nationals. In the context of rising unemployment concentrated among manual labourers and precarity caused by the pandemic, xenophobic sentiments were often expressed using a familiar refrain, as recounted by one individual:
*Miguel: [Costa Rican nationals] say, ‘Why don’t you look for work in your own country? Nicaragua is big,’ they say, ‘there is a place for you there, there you can wander around. You come here to Costa Rica to take away our jobs.’ … I tell them, ‘If you say I’m here to take your job, then why don’t you do it too? Look,” I tell them, ‘From what I understand, there is a scarcity of workers to pick coffee, cut sugar cane, gather melons, harvest oranges. ‘You can go do that job’, I say, ‘but you don’t like it because you’re used to doing jobs under shade. And we’re used to enduring sun, rain, cold, hunger, and that doesn’t make a dent in us’*

### 3.4. “There’s No Work, and I Have My Kids…”

In the face of this discrimination, finding stable work was a clear and enduring struggle. This struggle resulted in some participants being faced with the decision of choosing between unemployment, changing careers, or resorting to illegal work to sustain themselves. Even when work could be found, one participant described insecurity and the need to change careers:


*Miguel: My work has always been welding. But when one migrates, that has no value, it is worthless in another country. So, you have to make your way in life with whatever comes your way, that is, try to survive with what they offer you, the job you’re offered. As long as the work is honest. That is to say, many times along the way when you come here to migrate, there are always a few people who offer you jobs, but dirty jobs, better said, like dealing drugs or becoming a criminal. My idea was to work, and to work proudly.*


Women reported that difficulties finding work were compounded by childcare responsibilities, obstacles related to male partners, and sometimes needing to evade legal enforcement of prohibitions on informal work. Even when formal work opportunities could be found, further challenges related to English language requirements, minimum education levels, previous formal work experience, and limited employment opportunities remained. These pre-existing obstacles were felt even more acutely in the Costa Rica’s economically important tourism sector beset by a collapse in travel due to the pandemic. One participant described her experience trying to find employment in the tourism sector:
*Rosa: From the age of 15, yes, I started working, but with small shifts. Sometimes there were days when I cleaned windows, that is, cleaning a house, ironing clothes… To work in, like La Fortuna, you need experience and some knowledge of English, even if it is basic, in the hotels. Because look, even if you want to work as an attendant, they ask you for experience… I don’t understand why they ask for experience. I mean, I know how to grab a broom and a mop, clean the bathroom and-and, I mean, that’s something normal- in my opinion, we do that at home. I don’t understand why they ask you for experience.*

### 3.5. “You Have to Kill Yourself to Work”

If migrant workers were fortunate enough to find work, the challenges they faced were far from over. Participants reported long work hours, low pay, and exposure to pesticides and environmental hazards as being commonplace. Both formally and informally employed participants reported long working hours and physically demanding manual labour—particularly in the agricultural sector. One individual experienced these difficult working conditions from a young age:


*Patricia: When we first came here, I started working in a cassava plantation. I had never worked before. It was my first job, and I was just a little kid… There I started working peeling cassava. I peeled cassava. We worked until very late at night, from 6 [in the morning] to 12 at night or one in the morning. We worked very hard.*


In spite of these demanding conditions, migrant workers regularly reported being underpaid, and were often acutely aware that they were being paid lower wages than Costa Rican workers would accept for similar work. At the same time, these sub-standard Costa Rican wages were still substantially higher than what could be earned in Nicaragua, leading to a tension between gratefulness for an ability to afford commodities, such as a TV, and a sense of injustice in their remuneration in comparison to Costa Rican nationals. 


*Santiago: You know what a transnational company is. They come to this country, and they don’t come here to generate things that will benefit us; it is always going to benefit them. Because I work for [a transnational agricultural corporation]. I have worked for that company for more than 12 years. At first it was nice, it was good, I’m not going to complain, it was good. But [it eventually] resulted in lowering our economic level. Because my salary in that company is 10,600 Colones (US$16.50). That’s what I earn a day—10,600. You try buying something with 10,600. That doesn’t buy anything.*


Lastly, the most common exposures to occupational hazards were related to chemical and environmental risks. Agricultural workers reported repeated exposure to pesticides, including episodes that led to in the jury and death of coworkers. Exposure to dangerous chemicals was not limited to agricultural workers or unique to Costa Rica [21], as made clear by the death of a participant’s child following the fumigation of her house in Nicaragua. In addition to these chemical hazards, participants recalled having to lift heavy objects and working through hot, humid, and rainy weather. One participant recalled immediately noting these hazards’ impacts on his health:
*Miguel: So, when I came to work in a banana plantation, the first few days you get a flu, you get sick. Because those are jobs where you have to work in the rain, work in the water, work with rubber boots. And another thing is that the change in climate makes you feel different—especially because banana plantations use insecticides, poisons, fertilizers, and all that.*

### 3.6. “Get Your Insurance, They Say”

Of the many challenges migrant workers faced to access healthcare in Costa Rica, the cost and administrative requirements to obtaining health insurance were the most commonly cited. If documentation of legal residency could be obtained, participants reported a complex process for acquiring health insurance that involved high costs, multiple trips to government offices, long lines, and uncertainty over procedural requirements. One participant requiring medications to treat several chronic illnesses reported relying on garden-grown botanical medicines until their health insurance was approved:


*Corina: I’ve already been told by the social worker, I’ve been there, and she told me, ‘You have to live off an allowance, from what your daughters give you, and food. But it’s a pity,’ she tells me, ‘Unfortunately you don’t have your papers in order for us to help,’ and to this day they haven’t helped me because of that, because I don’t have my papers.*


As access to health insurance is mediated by access to formal work, the intersecting vulnerabilities faced by women and informal workers created additional burdens in accessing care. For participants who lost formal employment due to the pandemic, finding informal work could not replace the loss of associated health-insurance coverage. This lack of access to health services was said to limit access to preventive and outpatient care, and in some cases, resulted in exorbitant bills after accessing emergency care. One participant detailed how pandemic circumstances and familial changes had left them without access to health insurance:
*Patricia: In my case it has been a little difficult to get insurance, but it’s because you have to pay for it. In the past, since I have worked, I have always had my insurance, so I haven’t had any problem in the past. Afterwards when I didn’t work, my husband paid for insurance, so it was the same—I’ve always had it (insurance). Now with the pandemic, yes, I was left without insurance because my husband; well, we’ve separated, and he no longer pays. He stopped working because he’s getting older, so they don’t give him work.*

## 4. Discussion

This study amplifies and analyzes the voices of Nicaraguan migrants in Costa Rica to better understand the experiences of vulnerable workers in the context of the COVID-19 pandemic. Although participants regularly pointed to the negative impacts of the pandemic, these challenges were overwhelmingly the result of worsening pre-existing vulnerabilities related to unemployment, underemployment, discrimination, access to education and healthcare, and obtaining official documentation [22]. 

Although the pandemic may have featured more prominently in participant responses if research had been conducted later in the pandemic, the process of obtaining documentation was the theme that emerged most strongly as being at the root of vulnerabilities faced by migrants in Costa Rica. Internationally, the legal and social precarity faced by migrant agricultural workers and the resulting severe health consequences for this group have begun to receive more public attention [17,23,24,25]. Aligned with our findings, prior research indicated that legal status and access to services are intersecting vulnerabilities, and even if promised in theory, greater anticipation of the barriers faced by this group and greater alignment between policy and practice are required [24,26,27].

Using the themes emerging from our novel multimodal grounded approach, we propose a new model of the determinants of migrant health in Central America based on the organizing logic of the Dahlgren and Whitehead “rainbow model” in Figure 1 [28]. Inspired by the flag of the historic Federal Republic of Central America, a Phrygian cap representing liberty is seen over the peak of volcanoes representing the countries of Central America. Determinants are then ordered by individual-level factors; social and community networks; living and working conditions; and socio-economic, political, and environmental conditions. 

These determinants are further divided between the more-visible formal determinants in the rainbow on the top of the model, and more-hidden informal determinants underwater in the bottom of the model. As migrants undertake their journey from one country to another, they may experience simultaneous improvement and deterioration among various sets of these determinants; however, in keeping with participants’ determination and hope for a better future, we place the pursuit of wellbeing at the centre of the model.

Closest to this core are the more intrinsic determinants of wellbeing, such as agency, culture, and identity that migrants carry with them on their journey in pursuit of improved wellbeing. Beyond this core, home, family, and community provide further resources that are fundamental for wellbeing and provide the foundations for the pursuit of improvements in the areas of employment, education, and healthcare. 

While these can be partially found in the informal sector, the journey of migrants includes the pursuit of formal access to the goods and services provided by the State. The components that impede the pursuit of wellbeing include xenophobia, discrimination, and sexism at a more personal level, and obstacles to achieving formality at a more structural level, including bureaucratic, economic, and social determinants. In this model, we see that wellbeing is not only a goal that is sought with iterations of continued improvement but that a core of wellbeing is also the motor that allows for resilience in this pursuit.

One method of further understanding these emergent themes is through the lens of social citizenship—or the right of all members in society to shared minimum standards of living and economic security [29]. In fact, not only was entry into the Costa Rican health insurance and social security system desirable for the direct benefits of economic security and access to health services but it also became clear that these systems were seen as fundamental requirements for the full participation in Costa Rican society. Throughout Latin America, uneven but steady progress has been made in extending social citizenship rights through a combination of more broadly distributed economic growth and structural reforms of education and healthcare systems [30].

This case study demonstrates that, even in a country that is often pointed to as an exemplar of universal social protection, migrant workers continue to face exclusion due to administrative and financial barriers. This necessitates further investigation regarding the bureaucratic and navigational barriers that are faced by migrant workers and that are perhaps more easily made invisible within national contexts that espouse a comprehensive welfare system [31]. These systemic barriers to social inclusion have also been documented in countries of the global north, including Canada, the United States, and Europe [32,33,34].

Our data also point to the importance of including the inability to work and informal work—which accounts for 45% of Costa Rican workers—as important occupational risks among migrant populations [7]. Whether due to pandemic-related work stoppage, a lack of education or experience, inability to procure work permits, or xenophobic targeted layoffs; participants faced the prospect or reality of unemployment more regularly than their Costa Rican coworkers. This not only prevented migrant workers from accessing the aforementioned social services; it struck at the core values of honest work and self-sufficiency espoused by many participants. In the face of unemployment, migrants were faced with illicit employment options, relying on family for financial support, or finding some form of informal work. 

Although several participants did succeed in finding a way to financially support their family through informal employment, they continued to face long hours, hazardous work conditions, and lack of access to social security systems. Prior research also showed that migrant agricultural workers may be “perversely incentivized” to work beyond their physical limits because of their precarious and conditional status both in a country and in their workplace [35,36]. 

Given the very clear value of hard-work and self-sufficiency voiced by participants, structural workplace conditions may be able to exploit these qualities among this group. Labour unions may provide some reprieve for these migrants; however, their limited membership in society may also create further vulnerabilities even as individuals engage in self-advocacy and organizing to protect their health and safety [37,38].

Further exploration of understudied social positions held by migrant workers, such as Indigeneity, expressed in this study, also points to the need for further research to explore how such unique personal and collective histories may inform this populations’ access to services and general entry into society. Indigeneity as a point of analysis may be of particular relevance since prior research indicates that Indigenous populations may be overrepresented among Central American migrant workforces [39].

Gender also emerged as an important determinant of the challenges faced by and opportunities available to migrant workers. In addition to unequal access to formal education, sexism faced in workplaces, and unequal expectations of unpaid domestic labour, employment options were highly gendered, with agricultural labour often divided between planting and harvesting by men and the processing of agricultural products by women. 

Childcare emerged repeatedly as a source of financial liability for which some women became reliant upon financial assistance from current or former male partners. Echoes of these sources of economic insecurity faced by women migrant workers have been published from around the world; however, of particular interest to welfare state regime scientists is the persistence of these forms of gender-based insecurity in the context of a universalistic Latin American welfare regime [40,41,42].

## 5. Conclusions

Despite the many structural challenges facing migrants as they strove for social inclusion in Costa Rica, participants pointed to a sense of rootedness in a shared Nicaraguan identity as being a source of community strength and resilience. An ethic of hard work and independence most notably represented by the colloquialism of *pulseándola*, or pushing forward in the face of adversity, clearly drove many individuals to overcome seemingly insurmountable obstacles.

In times of need, mutual aid from family, friends, unions, and the Nicaraguan community allowed migrants to survive to fight another day. Prior research indicated that the presence of fellow compatriots, especially along a spectrum of settlement experiences, can provide important social ties and resources for migrant individuals and families [43]. A sense of belonging can be an important source of resilience for migrants who face economic, social, and political adversity [44,45]. 

Yet, looking further upstream, social determinants of health and wellbeing identified in this study, such as precarity and informality, dangerous working conditions, social and systemic discrimination, gendered occupational challenges, and access to legal residency and health insurance coverage are clearly systemic in nature and cannot be remedied through hard work and comradery alone. While there are many changes to national policy, living and working conditions, and society needs to address these multidimensional vulnerabilities in both the country of origin and host country, participants in this study prioritized access to legal documentation and social services as a primary and overriding concern. Clearly, these challenges are not limited to Central America. As we transition to a new phase of the COVID-19 pandemic, countries in both the global south and the global north must do more to provide migrants with a path to full social citizenship.

## Figures and Tables

**Figure 1 ijerph-19-06224-f001:**
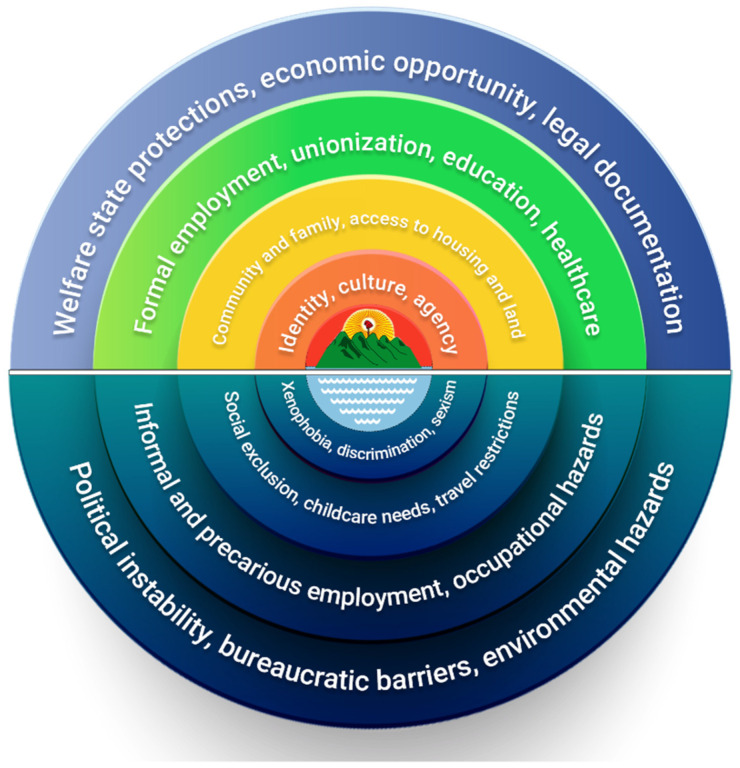
Determinants of migrant health in Central America. Based on the organizing logic of the Dahlgren and Whitehead “rainbow model” [28], the themes emerging from our novel multimodal grounded approach are ordered by individual-level factors; social and community networks; living and working conditions; and socio-economic, political, and environmental conditions. Our themes are further divided between formal determinants on the top of the model and informal determinants at the bottom of the model.

**Table 1 ijerph-19-06224-t001:** Working conditions, self-perceived health, prevention resources, and occupational health offices in Central America in 2018 (n = 9000) [3].

	Workers Exposed to Elevated Noise at Work		Workers Exposed to Nocive/Toxic Chemical Substances at Work		Workers Conducting Repetitive Movements at Work		Workers Who Must Work Very Quickly		Workers with Labor Risk-Prevention Services		Workers with Union Health and Safety Offices at Work		Workers with Poor Self-Perceived Health	
Country	Women	Men	Women	Men	Women	Men	Women	Men	Women	Men	Women	Men	Women	Men
Guatemala	18.4	17.1	7.6	27.7	51.4	67.9	38.6	44.9					19.2	24.3
El Salvador	26.3	29.1	6.9	36.1	69.4	73.9	54.0	50.5	7.8	6.7	6.6	7.8	31.6	26.7
Honduras	15.3	83.7	9.0	20.6	81.8	28.3	46.1	44.2	15.7	18.6	17.3	19.4	43.3	43.9
Nicaragua	11.8	73.8	8.9	20.9	68.2	26.8	35.5	36.4	23.1	29.1	20.7	25.3	47.9	43.4
Costa Rica	26.6	45.9	19.7	29.9	79.0	74.8	60.9	70.7	45.5	27.6	37.0	22.5	27.4	22.4
Panamá	21.0	29.6	20.1	27.2	47.8	62.8	41.2	40.5	40.6	41.3	31.1	36.3	30.2	23.1

**Table 2 ijerph-19-06224-t002:** Descriptive statistics of Central American countries and the three Costa Rican provinces within which the participants were recruited. The population, Gross National Income (GNI) per capita in thousands of US$ (2011 PPP), life expectancy, and Human Development Index (HDI) are presented for 2019 [18].

Country	Population	GNI per Capita	Life Expectancy	HDI
Costa Rica	5,047,561	$18,486	80.3	0.810
*San José province*	1,579,230	$19,324	-	0.828
*Alajuela province*	989,827	$18,736	-	0.804
*Limón province*	443,731	$16,017	-	0.768
Nicaragua	6,545,502	$5284	74.5	0.660
Belize	390,353	$6382	74.6	0.715
El Salvador	6,453,553	$8359	73.3	0.673
Guatemala	16,604,026	$8494	74.3	0.663
Honduras	9,746,117	$5308	75.3	0.635
Mexico	127,575,529	$19,160	75.0	0.779
Panama	4,246,439	$29,558	78.5	0.814

## Data Availability

The accompanying documentary film titled “More than Migrants” is available at https://www.youtube.com/watch?v=oDuBRoO6BHE (accessed on 5 February 2022). Additional video materials are archived through York University libraries and can be made available upon request.

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
