# Peer review of "Informality, Social Citizenship, and Wellbeing among Migrant Workers in Costa Rica in the Context of COVID-19"

_ijerph, 2022, doi:10.3390/ijerph19106224_

Round 1

Reviewer 1 Report

Comments  

The socio-economic crisis has and will inevitably continue to have a negative impact on people and will accentuate its imprint, producing negative consequences on the psychological and mental health of human beings. These problems will exacerbate the socio-economic crisis, reducing employment and increasing social inequalities. This article describes a dramatic human situation very focused on a migrant community like Nicaraguans. It is important to mention the fact that they are human beings fleeing the horrors of their land, where the greatest potential for future trauma lies in children. There is, for more in a pandemic situation, no sense of belonging. Women face more barriers in the health services, but the main one is the requirement, and rightly, to obtain a residence visa for a minimum legal stay in the territory of Costa Rica. In a disastrous environment in the country of origin and aggravated by the pandemic crisis that is still to be concluded, the best way for public authorities to resolve the opportunity of a lifetime is a policy of social inclusion. Even simple meals shared with friends and families help to integrate desperate people. This is the essential ingredient of the recipe, along with solidarity and tolerance. In practice, it is more difficult to integrate migrants from administratively disorganized countries, such as Nicaragua. However, accepting Nicaraguan migrants is a path full of barriers that still make it difficult for local companies and organizations to hire. If, on the one hand, the legal component is described as the great challenge for migrants, with all that this brings with it, on the other hand, civic education in Costa Rica is also essential to make the individual treatment given to them more dignified to every migrant. By the way, the authors speak of an organized public and health system in Costa Rica, but will the people who speak and deal with migrant issues be dedicated to fulfilling the Nicaraguan drama? (362-364).

Sugestions

Migrants have very high levels of unemployment and precariousness, but this is not the only case. First of all, training can be the key to opening doors to true inclusion. This is nice to say, but the beginning of the legalization process starts in the country of origin. Nationality and origin of migration starts in the country of origin, in this case Nicaragua. They should have made a reference to this question, but this is a suggestion for future studies on this phenomenon.

Another severe but very important point is that the migrants who arrive daily in Costa Rica are, even if instinctively, challenging the traditional image of Costa Rica. Not that this means, on the contrary, that Costa Ricans are inclined to shake off those who come from outside to invade their territory. Migrants have ages that make them potential competitors with the natives. It should be remembered that the unemployment rate in Costa Rica was 15.6% in January 2022 and 24% in July 2020 (Instituto Nacional de Estadística y Censos de Costa Rica), which are not low values. They do not allow much room for accommodation for more individuals coming from another country. Costa Rica's GDP can't take it, even though it grew by 10.3% in July 2021 alone!

Author Response

We thank the reviewer for their thoughtful response and agree that promoting policies of social inclusion in Nicaragua, Costa Rica, and throughout the Central American region is only way to sustainable promote wellbeing for migrants and citizens alike. We also acknowledge that by focusing on the current lived experience of Nicaraguan migrants in Costa Rica, our policy recommendations were primarily targeted to the Costa Rican government. We have now added additional information about the “push” factors driving migration from Nicaragua, as well as some brief comments and recommendations to address the need for policy change in Nicaragua. We must note that because study participants migrated from Nicaragua over the span of decades, we necessarily broadened our analysis of these push factors to those that pertained to long-term migration trends, rather than focusing on the contemporary political situation. Furthermore, in response to other reviewers, we have drawn parallels and policy recommendations for countries in the global North – including Canada and the United States – to make clear that social inclusion of migrants is very much a global issue requiring solidarity and bidirectional policy learning.

Reviewer 2 Report

Topic and methodology applied are very interesting and valuable. I would like to congratulate authors for the results and even more, for implementation of the entire research process

Good flow of the text, so the paper is reader-friendly.

I would like authors to consider the following:

  1. The title – especially regarding COVID’s part has not much reflection in the main body of the paper. However, the logic is present there – previous reasons of vulnerability are presented and then the COVID section is present. However, I cannot see efficient references to COVID in the conclusion part. I can suggest some developments/improvements of the conclusion section in order to bridge it more with the title.
  2. The conclusion part focuses a lot on community strength which I did not find as the core of the main part of the paper. I would suggest working on the conclusion part in order to draw conclusions based on the research result presented.
  3. I can understand that the authors do not start with theoretical framework while they apply the grounded theory. However, I think at the end of the paper there should be a kind of systematisation of findings – the theory should be formed out from the research results.

Author Response

We agree with the reviewer’s assessment that the previous version of the manuscript had a title that did not accurately reflect the content of the study and that the conclusion needed additional information to bring together the study’s findings. To address these issues, we have retitled the manuscript to “Informality, Social Citizenship, and Wellbeing Among Migrant Workers in Costa Rica” and have made considerable edits to the discussion and conclusion sections. As suggested by the reviewer, one of these changes systematizes our findings, putting forward a new model of the determinants of migrant health in Central America based on the Dahlgren and Whitehead “rainbow model” in Figure 1. We also expand our initial discussion of community strength to bring the focus back to the upstream determinants of health, provide high-level policy recommendations, and highlighting parallels between challenges faced by migrants in the Global North and Global South. We believe that these new additions have greatly improved the manuscript and thank the reviewer for their constructive suggestions.

Reviewer 3 Report

The manuscript uses a novel multimodal grounded approach synthesizing a documentary film, experiential education, and academic research to explore socioecological wellbeing among Nicaraguan migrant workers in Costa Rica during the Covid-19 pandemic. Particular emphasis is given to several challenges facing Nicaraguan migrants in Costa Rica which emerged from our grounded theory approach on topics ranging from precarity and informality, dangerous working conditions, social and systemic discrimination, and additional burdens faced by women. This is a very pertinent issue and the article deals well with this.

The fit between, theory and analysis is well developed, and aids the authors in his/her/their analysis, and in framing the conclusion. 

The theoretical part is well developed. A clear stating and focusing of the argument is provided. Results are linked suitably to the other sections of the article. A well-organized and compelling discussion section is provided as well.

Methods are appropriate and the fit between theoretical discussion and methodology is well formulated. 

The results are of interest for practice, health and migration policy and society more generally. 

The article highlights the central literature concerning the theme. 

The manuscript explores an interesting and original topic. It offers a robust theoretical frame and a well-organized and convincing discussion of findings. The authors has/have paid attention to the clarity of expression and readability, such as sentence structure. The quality of communication is good.

The authors address a significant research subject and presents interesting field material. This article could be a starting point for further research.

Minor suggestion:

A brief table with social and demographic data from the sample could have used in order to provide the reader(s) with further information. 

Author Response

We thank the reviewer for their thorough and generous comments on the value of this study. We agree that it is a pertinent issue not only for Costa Rica and Central America, but for countries in the Global North as well. We have followed the reviewer’s helpful advice to add a table with social and demographic data, but instead of focusing on our participants, we have presented population, Gross National Income (GNI) per capita in thousands of US$ (2011 PPP), life expectancy, and Human Development Index (HDI) for the provinces in which participants live and countries in the region to further contextualize our results in Table 2. With the addition of this new information, we hope that our results will inspire further research and contribute to informing policy change.

Reviewer 4 Report

The study methodology is indeed novel. I only wish that more participants had been included.

The emergent themes are relevant and illustrative of the challenges faced by migrant workers. The authors may wish to connect these findings to research in other countries that rely on migrant labor, such as the United States, where migrant workers are similarly marginalized.

Author Response

We have taken the review’s helpful advice to connect our findings to research in other countries that rely on migrant labor such as the United States and Canada in the introduction (line 84), discussion (line 426), and conclusion (line 486). Although we may have captured a larger diversity of experiences by including more participants, we are confident that our in-depth exploration of participants’ experience produced data with enough richness to inform our analysis and subsequent discussion. We also aim to produce further research delving into these issues in the future and hope our study encourages other researchers to do so as well.

Reviewer 5 Report

Dear Editors,

Thank you for sending me this article. And thank you to the authors for submitting such a polished and clear paper. I particularly enjoyed the extensive use of direct quotations and the authors’ commitment to foregrounding workers’ voices. This shone through in the analysis, where the authors handle the complexities of workers’ lives and stories with care and nuance. The article is well organized, clearly written, and makes a valuable contribution to the literature on migrant work regimes and social wellbeing. The article requires some clarification as to the place of COVID-19 in the argument, the multi-modal approach, and the literature to which the authors hope to contribute. These changes should be straightforward and easy to make but, because they pertain to the broad framing of the paper, my recommendation is that that the authors make these changes and resubmit the paper for consideration.

This article investigates the lives and wellbeing of Nicaraguan migrant workers in Costa Rica, with a particular focus on the ways this group of workers has been impacted by the COVID-19 pandemic. The authors use what they call a multimodal grounded approach—combining interviews, documentary film, and opportunities for feedback and collaboration—that aims to empower workers to tell their own stories and participate in the production of knowledge that flows from this research. The authors argue that, rather than see the pandemic as an immediate health threat, workers have more acutely experienced its impact in the ways it has intensified intersecting vulnerabilities already facing these workers, spanning precarity, dangerous working conditions, discrimination, barriers to accessing documentation and insurance, and lack of childcare and social supports. The article contributes to the literature on migrant labour regimes and social wellbeing through a detailed study of the Costa Rican context. In particular, the article makes the valuable contribute of documenting the persistence of migrant worker insecurity despite Costa Rica’s acclaimed welfare state protections.

My first concern has to do with how the COVID-19 pandemic figures in the article. While COVID-19 features prominently in the title of the paper, the reality of the pandemic feels somewhat tacked on to the rest of the paper. For example, the pandemic is rarely explicitly mentioned in the results section until section 3.6 and is not mentioned in the paper’s conclusion. I would suggest that the authors either integrate the context and impacts of the pandemic into the paper or change the title of the article to take the emphasis off COVID-19.

I appreciate that the authors’ want to illustrate that the pandemic has only deepened pre-existing precarities facing these workers and is not, as such, their primary health concern (or even a concern primarily about health). I think this is an important finding and am not suggesting the authors’ take the focus off this reality. Rather, if the authors want to keep the focus on COVID-19 in the paper’s framing, I think they could add more detail as to how the pandemic has worsened these challenges and inequities. One idea would be to thread this theme through the individual subsections in Section 3. Rather than reserve the theme of COVID-19 for the final subsection, the authors could consider foregrounding the results section with the point they make in 3.6 and then detailing in the rest of the subsections specifically how COVID-19 has exacerbated these dynamics.

If the authors opt for this direction of more meaningfully integrating the pandemic context, I would also suggest more overtly framing the paper in research that addresses this theme. The authors do cite two studies that specifically focus on COVID-19 and migrant workers (line 87, refs 6 and 7) but I wonder if the authors could broaden their engagement with literature in this area, whether focused on migrant workers or on social wellbeing, inequality, work, and COVID-19 more generally.

In addition to helping to clarify the place of COVID-19 in the analysis, this would also help clarify the paper’s contribution to the literature more generally. As it stands, I found it somewhat difficult to discern how the authors wish to situate the paper in the literature, beyond the geographic specificity of the study’s focus on Costa Rica. Related, the element of Costa Rica’s welfare state (which is emphasized nicely in the discussion) is an interesting detail that could be highlighted more prominently in the introduction and literature review. I have the sense that this is one of the paper’s unique contributions, but this could be more clearly stated.

My second area of concern relates to what the authors call their “multimodal grounded approach” to this research. I think the paper simply requires some clarification as to what this means. In the abstract, the authors reference synthesizing documentary film, experiential education, and academic research. In the Data Analysis section, they offer a nice detailed description of how they analysed their data, including through collaboration between students and professors at universities in Costa Rica and Canada and through the production and screening of a documentary film. Can the authors provide more detail as to how these other modalities relate to the interviews? For example, what came out of the film screenings? Did these shape the analysis? If so, how? If not, what role did they play in the broader methodological approach? The authors might also consider including some reflections on the limitations of these collaborative methods (would workers have felt comfortable sharing critical feedback?) and how they worked with and accounted for these limitations.

There are also some specific references that need explanation. What is the Grounded project? What is its relation to the research in this specific article? I also think the concept of “emergent grounded theory” (line 148) merits some explanation, including in the specific context of this research.

Finally, I have a few specific comments:

  • In the introduction (beginning on line 94) the authors explain that, over the last decade, migrant workers’ access to government services has been declining while exposure to risks and hazards is increasing. What is the broader context for this shift? Is this shift in Costa Rica just part of the broader rise of migrant precarity in Central America, or is something specific happening in Costa Rica?
  • What is the Grounded project and how does this project fit into it (lines 93, 140)?
  • There is no mention of COVID-19 in the explanation of the study’s aims (beginning line 94). If the authors retain COVID as central to the paper’s framing, consider including it here.
  • In section 2.1 (Data collection) there is no mention of the multimodal approach. Please explain the multimodal component here and how it relates to the interviews. If the multimodal component is contained to the analysis, it would help to simply indicate that there.
  • I would be interested to hear more about the production of the documentary (line 146) as part of the research methods, if relevant, rather than as simply part of the analysis (as it is currently explained in Section 2.2.
  • There is no mention of COVID-19 in the conclusion. If the authors retain COVID as central to the paper’s framing, it should figure in the conclusion.

Author Response

We thank the reviewer for their detailed and highly constructive comments, which we believe have significantly improved the manuscript.

With regards to the framing of the paper around the COVID-19 pandemic, we agree that there was a notable imbalance between emphasis in the first half of the manuscript (as well as the title) and the lack of attention paid to the pandemic after the presentation of the results. This can be partially explained by the fact that the pandemic and its impact on Nicaraguan migrants motivated the conceptualization of this research project, which is important background information that is now noted in the introduction and methods. Despite this initial motivation, and as the reviewer rightly notes, participants have more acutely experienced the pandemic’s impacts in the ways it has intensified intersecting and pre-existing vulnerabilities.

In keeping with our grounded approach, we have decided to de-emphasize the importance of the pandemic in the title, we have reorganized the section focused on the pandemic (previously section 3.6) to the beginning of the results section (now section 3.1) to emphasize the pandemic as our port of entry into these intersecting vulnerabilities, and we have expanded our discussion of the pandemic’s impacts in Costa Rica and beyond in the discussion section. Although the reviewer provides excellent advice on further ways to reorient our manuscript using relevant literature on COVID-19 and its impacts on migrant workers, we feel that a more extensive change in the focus of our discussion would have prioritized our research motivation over the data that emerged from the interviews themselves.

With regards to our contribution to literature, we do believe that our analysis of the mismatch between Costa Rica’s lauded welfare state and the reality facing Nicaraguan migrants in the country is an important takeaway that can be generalized to other contexts. We have attempted to expand on this point throughout the discussion. We have also attempted to expand on the significance and limitations associated with our methodological approach in several areas. First, we have removed the method from the title, as we decided that its uniqueness was not central to our findings or contributions. We have also added further details on the questions posed in lines 139 and throughout section 2.2 and reflected on limitations in line 201. Finally, we have added a new systematisation of findings, putting forward a model of the determinants of migrant health in Central America based on the Dahlgren and Whitehead “rainbow model” in Figure 1, which we believe is another novel contribution to literature.

Finally, in response to minor comments, we have added a description of the Grounded Project (lines 110); removed mention of “emergent grounded theory” which misleadingly combined two distinct approaches/concepts; and added more detail to the description of our multimodal approach and production of the documentary in Sections 2.1-2.2. We are confident that the manuscript has been significantly strengthened by these edits and thank the reviewer for taking the time to provide us with this constructive review.

Round 2

Reviewer 2 Report

I appreciate the improvements made after the first revision round. I think that now the paper is more coherent, scientifically sound and can be published.

Once  again congratulations on the empiricial research.

Author Response

We thank the reviewer for their support in improving our original manuscript into a more easily understood and scientifically sound contribution to the literature.

Reviewer 5 Report

Dear Editors,

The authors have made significant changes to the article, which have nicely addressed all of my major concerns. The revised article is clear and persuasive. I look forward to seeing it published.

I just have a few small outstanding comments:

  • The authors have moved their discussion of COVID-19 in the results section to the beginning of the section (3.1, line 207), which works nicely. However, it seems they forgot to remove the old section 3.6 (line 380). Moreover, I would still appreciate hearing some detail included in each subsection of the results section as to how specifically the pandemic has intersected with each specific challenge. This is a comment I made in my original review, so perhaps the authors have decided it simply doesn’t work with the structure of the paper or that their results don’t reflect this way of thinking about these challenges. If the authors feel there are concise and specific ways they could describe how the pandemic has worsened these challenges, I still think that would strengthen the paper.
  • The rainbow model on page 10 is a nice addition. As someone who is not familiar with this model I had a hard time understanding if the content inside the model was developed by the authors or whether this is inherent to the original rainbow model. If the content was developed by the authors, or if the image here is a modification of the original rainbow model, simply clarify this and explain what you have changed. If the content is inherent to the original rainbow model, please just explain the contribution the character of the “new model” the authors are putting forward.
  • I like the new title but also think that, given the extent to which the authors have added content and context for the place of the pandemic in the article, they could now include it in the title if they so desired (e.g., Informality, Social Citizenship, and Wellbeing Among Migrant Workers in Costa Rica in the Context of COVID-19). This would also bring the title into line with the content of the abstract as it is currently written. If the authors choose to keep the current title, they may want to make some small changes to the abstract to reflect their de-emphasis on the pandemic in the paper generally.

Author Response

We are grateful for the reviewer’s notes and suggestions on ways to improve our manuscript, and we are very pleased to have addressed all of their major concerns. We also appreciate the three additional minor comments provided, which have served as an opportunity for a final review for coherence following the substantial changes to the original manuscript. For more details on the specific points raised by the reviewer:

We agree with the reviewer’s suggestion to identify concise and specific ways that the pandemic has worsened pre-existing challenges faced by participants and have now ensured that each subsection of the findings details at least one way that the pandemic has intersected with each specific challenge. With the rearranging of the section titled “COVID has come to mark us all” to the start of the findings section, these newly detailed intersections strengthen the cohesiveness of the presentation of findings. With regard to subsection numbering, we have ensured that subheadings are sequentially numbered from 3.1 (“COVID has come to mark us all”) to 3.6 (“Get your insurance, they say”).

The reviewer’s helpful suggestion to provide clarity on whether we recreated the Dahlgren and Whitehead “rainbow model” or if we have created a truly original model has prompted us to clarify that with the exception of the organizing logic (individual-level factors to socio-economic, political, and environmental conditions), this is an original contribution to literature based on our study’s findings. This is now made clearer in the discussion and figure legend, and we have added a new paragraph detailing the determinants of health and wellbeing contained in the model.

Now that we have thoroughly integrated content related to the pandemic throughout the manuscript, we also agree with the reviewer’s suggestion to add COVID back to the manuscript title. It is now titled “Informality, Social Citizenship, and Wellbeing Among Migrant Workers in Costa Rica in the Context of COVID-19”